# BINARY DIFF SUMMARIZATION USING LARGE LANGUAGE MODELS

## ABSTRACT

Security of software supply chains is necessary to ensure that software updates do not contain maliciously injected code or introduce vulnerabilities that may compromise the integrity of critical infrastructure. Verifying the integrity of software updates involves binary differential analysis (binary diffing) to highlight the changes between two binary versions by incorporating binary analysis and reverse engineering. Large language models (LLMs) have been applied to binary analysis to augment traditional tools by producing natural language summaries that cybersecurity experts can grasp for further analysis. Combining LLM-based binary code summarization with binary diffing can improve the LLM's focus on critical changes and enable complex tasks such as automated malware detection. To address this, we propose a novel framework for binary diff summarization using LLMs. We introduce a novel *functional sensitivity score* (FSS) that helps with automated triage of sensitive binary functions for downstream detection tasks. We create a *software supply chain security* benchmark by injecting 3 different malware into 6 open-source projects which generates 104 versions, 392 binary diffs, and 46,023 functions. On this, our framework achieves a precision of 0.95 and recall of 0.71 for malware detection, displaying high accuracy with low false positives. We outperform an existing industry-style rule-based baselines by $\approx 4\times$ higher recall on malware detection while maintaining high precision. Across malicious and benign functions, we achieve FSS separation of 3.0 points, confirming that FSS categorization can classify sensitive functions. We conduct a case study on the real-world XZ utils supply chain attack; our framework correctly detects the injected backdoor functions with high FSS.

## 1 INTRODUCTION

Binary analysis is fundamental to cybersecurity, enabling critical tasks like vulnerability discovery, malware analysis, and software supply chain integrity. Reverse engineering low-level binaries to extract high-level functionalities is essential for understanding their behavior without access to source code (Cifuentes, 1994; Schulte et al., 2018; Shoshitaishvili et al., 2016). Binary differential analysis (binary diffing) extends binary analysis by comparing two versions of a binary to understand what has changed, allowing analysts to focus on the modifications to find newly introduced bugs or security flaws. This approach is especially relevant for software supply chain security that involves verifying the integrity of software updates (Reichert & Obelheiro, 2024). Binary diffing supports critical security tasks like identifying patched vulnerabilities (Brumley et al., 2006), clustering malware variants (Royal et al., 2006), detecting vulnerabilities in binary distributions (Zhao et al., 2022). Malicious actors may inject hidden code into a program binary or into the source code of dependent open-source libraries, leading to devastating downstream impact as exemplified by recent incidents such as 3CX (FortiGuard Labs, 2023), SolarWinds, log4j, and XZ utils (Williams et al., 2025). These concerns are magnified in the context of the embedded systems supply chain. Unlike enterprise software, embedded devices are deployed in remote or inaccessible locations. Software updates require significant effort. Due to this, corrupted or compromised updates may persist for extended periods. Embedded firmware is distributed as monolithic binaries which incorporate several projects into one blob, making verification difficult (Shirani et al., 2017). This necessitates initial verification of software integrity before deployment via binary analysis and reverse engineering.

Reverse engineering is an inexact process, hence binary analysis tools often recover source code in an obscure format, requiring significant effort and domain expertise to understand (Cao et al., 2024). Machine learning (ML) and large language models (LLMs) have been used to improve the reverse engineering output quality, such as by predicting variable names and types (Lacomis et al., 2019; Nitin et al., 2021), decompiling with translation ML models (Armengol-Estapé et al., 2024; Udeshi et al., 2025), and binary code summarization with LLMs (Jin et al., 2023; Tan et al., 2024).

Binary diffing tools are built on top of binary analysis methods and hence face similar issues of obscurity, hard-to-understand outputs. Current tools reliably identify modified binary components by employing binary code similarity metrics; however, cybersecurity experts require significant effort to understand the code changes to identify vulnerabilities or detect malicious injected code. We propose *binary diff summarization* to augment binary diffs with natural language summaries produced by an LLM. Additionally, we introduce the *functional sensitivity score* (FSS), a novel categorization method to triage binary functions such that sensitive behaviors that reveal vulnerabilities or malware are marked with a high score. We evaluate the binary diff summarization and functional sensitivity score for the software supply chain security task of detecting malware injected into open-source programs. For this, we construct a benchmark by injecting malware into multiple versions of open-source programs to construct compromised software updates across clean/injected versions.

The contributions of this paper are threefold: (i) A novel framework for *binary diff summarization* that augments outputs from binary diffing tools with LLM-generated natural language summaries for improved code understanding; (ii) The *functional sensitivity score*, a novel method to triage sensitive function behaviors that highlight vulnerabilities and malicious code; (iii) A *software supply chain security benchmark* of open-source programs injected with 3 different malware, comprising of 6 projects, 104 binary versions, 392 binary diffs, and 46,023 functions.

## 2 BACKGROUND AND RELATED WORK

**Binary Differential Analysis:** Binary differential analysis (binary diffing) is the process of identifying changes between compiled binaries at different granularities, such as instructions, basic blocks, or complete binary formats (Haq & Caballero, 2021). Unlike source-level diffing, which benefits from static code analysis, binary diffing is considerably more challenging due to compiler optimizations, instruction set variations, and obfuscation techniques (Linn & Debray, 2003). Early efforts such as BinDiff (Flake, 2004), DarunGrim (Oh, 2008), and BMAT (Wang et al., 2000) relied on syntactic and graph-based similarity across control-flow graphs (CFGs), with later improvements addressing register allocation and instruction reordering (Dullien & Rolles, 2005). Subsequent works have expanded these ideas. Diaphora leverages SQL-based heuristics for CFG matching (Koret, 2015–2025), while Asm2Vec (Ding et al., 2019) introduced function embeddings resilient to compiler optimizations. More recently, deep learning methods such as jTrans (Wang et al., 2022) incorporated transformers with jump and control flow awareness. QBinDiff (Cohen et al., 2024) reframed diffing as a graph-alignment problem, achieving robustness against obfuscation as shown in the evaluation of Cohen et al. (Cohen et al., 2025). Other approaches, including Binhunt (Gao et al., 2008), and BinSlayer (Bourquin et al., 2013) employed symbolic execution, graph isomorphism and unsupervised learning to capture semantic differences more precisely. These approaches primarily tackle the challenge of binary code similarity matching which extends to binary diffing. To our knowledge, existing works do not directly target malware-injected software updates. A recent tool `malcontent` ChainGuard (2025) uses binary diffing to augment a rule-based malware scanner. However, rule-based scanners require frequent updates to their rules otherwise they are brittle to novel malware. We utilize `malcontent` as a baseline. Appendix A.1 provides more details of the related approaches.

**LLMs for Binary Analysis:** Jin et al. (2023) introduced BinSum, a benchmark of over 557K binary functions across multiple architectures and optimization levels, along with novel prompt optimization strategies and semantic evaluation metrics for binary summarization. Dil et al. (2025) applied LLM-guided prompting to filter noisy vulnerability patch data in the BigVul dataset (Fan et al., 2020), improving the accuracy of downstream vulnerability prediction models, while Yu (2025) proposed DeepDiff, which embeds decompiled functions for similarity search and combines control and data flow analysis to detect logic-altering changes

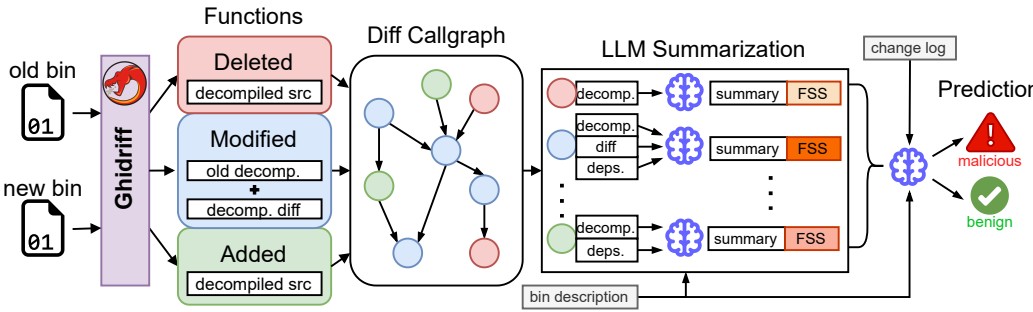

Figure 1: Overview of the binary diff summarization framework. Ghidriff provides *added*, *modified*, and *deleted* functions that are merged into a diff callgraph. Each function undergoes LLM summarization and FSS classification. Finally, prediction happens to label the diff as malicious or benign.

in binaries. Shang et al. (2024) constructed a benchmark for reverse engineering tasks such as function name recovery and summarization, systematically evaluating LLM capabilities. Hussain et al. (2025) developed Vul-BinLLM, which augments decompilation with contextual vulnerability annotations and employs in-context learning, chain-of-thought prompting, and memory management to improve detection accuracy. Lin & Mohaisen (2025) systematically evaluated LLMs for vulnerability detection in Java and C/C++ programs, highlighting cross-language performance, prompting strategies, and configuration best practices. Wong et al. (2023) explored recompilable decompilation, proposing a hybrid two-stage approach where LLMs correct syntax errors in decompiled outputs and resolve runtime memory errors, enabling regenerated executables that preserve original functionality. Chen et al. (2025) introduced ReCopilot, an expert LLM for binary analysis that integrates variable data-flow and call-graph information with test-time scaling, and through continued pretraining, supervised fine-tuning, and direct preference optimization, achieved up to a 13% improvement over state-of-the-art models in function name recovery and variable type inference.

## 3 METHOD

Figure 1 is an overview of the binary diff summarization pipeline. In the context of software supply chain security, the pipeline concludes by producing a malicious/benign prediction. The summaries and FSS scores can be used for tasks such as vulnerability detection or patch identification.

**Ghidriff:** The pipeline begins by taking two binaries namely *old* and *new*. We use Ghidriff (McIntosh, 2023) as the binary diff tool. Ghidriff uses the Ghidra decompilation engine to perform the initial analysis of both binaries, then computes corre-

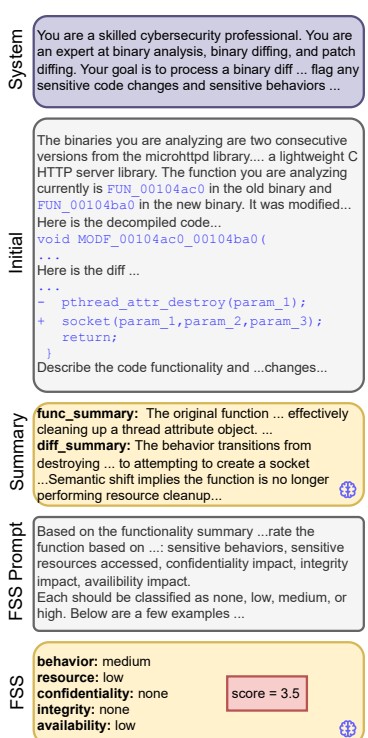

Figure 2: Example of LLM summarization and FSS for a modified function from microhttpd.

lations across functions from *old* and *new*. The correlation reveals whether a pair of functions match exactly, match approximately, or do not match. Ghidriff outputs three lists of functions: *deleted* contains functions present in *old* that do not match with any function in *new*, *added* contains functions present in *new* that do not match with any function in *old*, and *modified* contains functions that match approximately. Thus, functions that match exactly are removed from the diff, so only the binary changes remain. As the symbols in both binaries are stripped, the *modified* functions will show up with different names in the decompiled code depending on their hexadecimal address. We rename the *modified* functions to a consistent name incorporating both the old and new address, and update all referenced locations, so that this name difference does not show up unnecessarily.

**Diff Callgraph:** For the functions in the three lists, we extract the decompiled source code from Ghidra analysis. For *modified* functions, we additionally compute a textual diff of the decompiled code using the Python `difflib`[1] module to provide a succinct representation of the changes. The LLM summarization happens function by function and some information about the function dependencies (in terms of other functions it calls) needs to be provided for the LLM to understand the functionality correctly. This dependency information is captured in the diff callgraph. The diff callgraph is essentially the merged callgraph of the *old* and *new* binaries, where only the *added*, *deleted*, and *modified* functions are preserved. Instead of merging and trimming down the full callgraphs of the binaries, we construct the diff callgraph by directly analyzing referenced dependencies in the three diff function lists.

**LLM Summarization:** The functions are processed in a reverse breadth-first traversal starting from leaf nodes of the diff callgraph to ensure that a function's dependencies are processed before it. For each function, the decompiled source code along with summaries of the dependencies are passed to the LLM. For *modified* functions, the textual diff of decompiled code between *old* and *new* is also passed. The summary and FSS are generated via two separate prompts. The first prompt asks for a functionality summary and an optional diff summary. The second prompt continues the conversation (the LLM sees its previous output) and asks for the FSS. Figure 2 shows an example conversation for a modified function from `microhttpd`, where the LLM first correctly identifies the changed functionality and then proceeds to mark the FSS categories appropriately.

**Functional Sensitivity Score:** The FSS design is inspired by the common vulnerability scoring system (CVSS) (Howland, 2022) such that functions of interest can be marked during binary analysis using consistent categories. CVSS helps score the severity of vulnerabilities after they are identified with distinct categories and classification options, for example attack complexity (low, high) and privileges required (none, user, administrator). This allows for better vulnerability classification by cybersecurity professionals than picking abstract numerical values. CVSS aggregates the category classifications into a severity score from 0 to 10. CVSS does not directly apply for vulnerability or malware detection. Thus we design FSS with similar goals to provide meaningful categories and classifications for scoring functional sensitivity. We pick five categories with examples:

- **Sensitive behaviors** ($B$): reading system info, opening sockets, forking processes
- **Sensitive resources** ($R$): network, system files, hardware devices
- **Confidentiality impact** ($C$): sending files over network, reading passwords or keys
- **Integrity impact** ($I$): modifying system configuration, overwriting files, encrypting data
- **Availability impact** ($A$): disabling system services, consuming unnecessary resources

Each category is classified as none, low, medium, or high. We provide examples to the LLM of each category and each classification to ground its outputs. These examples can be adapted to different scenarios and environments to better guide the LLM.

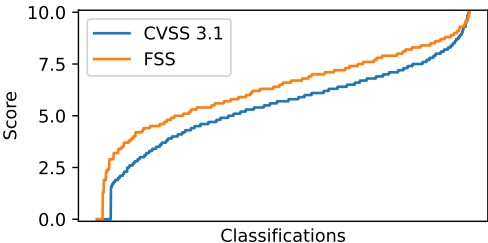
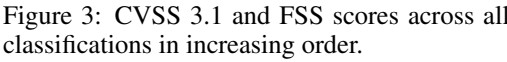

Figure 3: CVSS 3.1 and FSS scores across all classifications in increasing order.

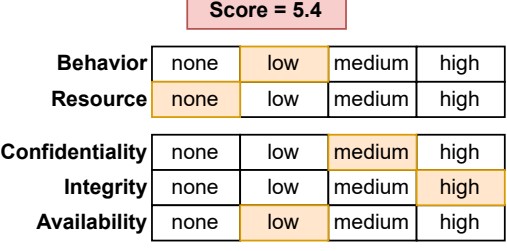

Figure 4: Example of an FSS classification with the aggregate score.

---

[1] https://docs.python.org/3/library/difflib.html

Similar to CVSS 3.1, the final score is aggregated using the formula:

$$S = 1 - (1 - B)(1 - R)$$
$$M = 1 - (1 - C)(1 - I)(1 - A)$$
$$FSS = \begin{cases} \text{roundup}(5.3S + 6.1M) & M > 0 \\ 0 & \text{otherwise} \end{cases}$$

where $B, R, C, I, A$ are as defined above, $S$ is sensitivity aggregate, $M$ is impact aggregate, and roundup rounds up values to one decimal place. Weights for $B$ and $R$ are $\{\text{none} = 0, \text{low} = 0.1, \text{medium} = 0.35, \text{high} = 0.6\}$, while weights for $C$, $I$, and $A$ are $\{\text{none} = 0, \text{low} = 0.22, \text{medium} = 0.39, \text{high} = 0.56\}$. Equations for $S$ and $M$ are structured to produce a high score when any one of the components are marked higher, similar to CVSS 3.1. The weights and coefficients were tuned such that FSS captures the scores from 0 to 10. Figure 3 shows the scores of all classifications in increasing order for CVSS 3.1 and FSS, demonstrating that FSS behaves similarly to the industry-standard CVSS 3.1. Figure 4 shows an example classification and its score.

**Prediction:** The last step of the pipeline is the prediction that outputs whether the summarized diff contents resemble malicious injection or a benign software update. We implement this step by passing the top $k$ functions with highest FSS to the LLM and prompt it to output either MALICIOUS or BENIGN by reasoning about whether the changes match the project description.

## 4 EVALUATION

### 4.1 SUPPLY CHAIN SECURITY BENCHMARK

We construct a benchmark for software supply chain security by picking six popular open-source projects spanning command line utilities and libraries `gzip`, `openssl`, `tar`, `sqlite`, `microhttpd`, and `paho-mqtt`. Additional details are provided in Appendix A.2. Table 1 shows the project description, number of versions selected, number of diffs from taking consecutive version pairs, and total number of *added*, *deleted*, and *modified* functions across all diffs. Each project is compiled in a Ubuntu 20.04 docker container with the default compiler GCC 9.4.0. Pairs of binaries of consecutive versions are treated as software updates and we use them for binary diffing. We collected 104 versions across the 6 projects, generating 98 software update pairs.

| Project | Description | Versions | Diffs | Functions |
|---------|-------------|----------|-------|-----------|
| `gzip` | File compression utility | 5 | 16 | 1209 |
| `openssl` | Cryptography library and utility | 29 | 112 | 9722 |
| `tar` | File and directory archival utility | 10 | 36 | 8936 |
| `sqlite` | Single-file SQL database library | 28 | 108 | 16682 |
| `microhttpd` | Lightweight HTTP server library | 23 | 88 | 6648 |
| `paho-mqtt` | MQTT lightweight messaging library | 9 | 32 | 2826 |
| **Total** | | 104 | 392 | 46023 |

Table 1: Details of software supply chain security benchmark.

Additionally, we implement three malwares to inject into the source code of each project. Details of each malware are provided in Appendix A.3.

- `rware`: a ransomware (Li, 2021) that encrypts user files using AES and ECDH encryption
- `rat`: a remote access trojan (Kara & Aydos, 2019) that initiates a reverse shell with remote server
- `botnet`: a bot network (Antonakakis et al., 2017) for denial-of-service attacks on servers

Each malware is implemented as C code contained in one source file that is copied into the project source directory and added to the build system. The entry point of each malware is a C function that takes no arguments and returns no values. For each project, we determine a trigger point that is not reachable in normal operation of the project but can be triggered by the attacker with specific mal-formed configurations, for example, passing an attacker-defined command line option. We obtain 4 binary diffs per version pair by considering a software update from a clean binary of the former version to the clean and injected binaries of the latter version. In total, this makes 392 diffs.

| Program | $k = 5$ | | $k = 10$ | | $k = 5$ w/ change | | $k = 10$ w/ change | |
|---|---|---|---|---|---|---|---|---|
| | $P$ | $R$ | $P$ | $R$ | $P$ | $R$ | $P$ | $R$ |
| Overall | 0.95 | 0.71 | 0.94 | 0.63 | 0.96 | 0.70 | 0.94 | 0.62 |
| gzip | 1.00 | 0.83 | 1.00 | 0.67 | 1.00 | 0.83 | 1.00 | 0.83 |
| openssl | 0.88 | 0.51 | 0.85 | 0.49 | 0.88 | 0.55 | 0.87 | 0.54 |
| tar | 0.88 | 0.56 | 0.84 | 0.59 | 0.88 | 0.56 | 0.88 | 0.56 |
| sqlite | 0.98 | 0.79 | 1.00 | 0.57 | 1.00 | 0.76 | 1.00 | 0.56 |
| microhttpd | 0.96 | 0.83 | 0.98 | 0.79 | 0.98 | 0.79 | 0.96 | 0.76 |
| paho-mqtt | 1.00 | 0.92 | 0.96 | 0.92 | 1.00 | 0.88 | 1.00 | 0.75 |

Table 2: *GPT5 mini* malware detection precision ($P$) and recall ($R$) across programs. $k$ refers to how many functions with highest score are provided for prediction step. "w/ change" refers to program changelog being provided for the prediction step.

## 4.2 METRICS

**Malware Detection:** Prediction output is evaluated against ground truth labels for each diff. Diffs with binaries containing the injected malware are labeled MALICIOUS and diffs with clean binaries are labeled BENIGN. Treating the MALICIOUS label as positive, we compute precision and recall to evaluate accuracy of malware detection. False positives would be clean diffs labeled as MALICIOUS. False negatives would be diffs with injected malware labeled as BENIGN.

**FSS Separation:** It is difficult to evaluate the quality of FSS scores assigned by an LLM without human-labeled scores for functions in the diff. Even in clean diffs, functions may show different behaviors and thus different FSS. In our benchmark, we mark functions from the original code as benign and injected functions as malicious. FSS scores are averaged as $FSS_{\mathrm{ben}}$ and $FSS_{\mathrm{mal}}$ across a binary. Their distributions are checked to see if malicious functions score higher than benign ones. Higher separation of the distributions of $FSS_{\mathrm{ben}}$ and $FSS_{\mathrm{mal}}$ will indicate better FSS quality.

## 5 RESULTS

**Experimental Setup:** We evaluate with two commercial LLMs, *GPT5 mini* and *GPT5 nano* (OpenAI, 2025a), and three open-source LLMs, *GPT OSS 20B* (OpenAI, 2025b), *Qwen3 30B*, and *Qwen3 8B* (Qwen, 2025). All five LLMs are run in thinking/reasoning mode. Reasoning effort is set to "low" for *GPT5 mini*, *nano*, and *GPT OSS 20B* models. *Qwen3 30B*, *8B*, and *GPT OSS 20B* models are run via Ollama on a server with two NVidia L40 GPUs. The models are run with default hyperparameter settings as follows: temperature of 1.0 and top-$p$ of 1.0 for *GPT5 mini*, *GPT5 nano*, and *GPT OSS 20B*; temperature of 0.6 and top-$p$ of 0.95 for *Qwen3 8B* and *Qwen3 30B*. We do not evaluate the highest capability *GPT5* on the full benchmark due to high API costs and because the smaller models suffice as seen in the results. Appendix A.4 plots token and cost analysis.

| Summ. | Pred. | $P$ | $R$ |
|---|---|---|---|
| *GPT5 mini* | *GPT5 mini* | 0.95 | 0.71 |
| *GPT5 nano* | *GPT5 nano* | 0.85 | 0.35 |
| *GPT OSS 20B* | *GPT OSS 20B* | 0.93 | 0.42 |
| *GPT5 mini* | *GPT5* | 0.97 | **0.72** |
| *GPT5 nano* | *GPT5 mini* | 0.90 | 0.50 |
| *GPT OSS 20B* | *GPT5 mini* | 0.96 | 0.63 |
| *Qwen3 30B* | *GPT5 mini* | 0.99 | 0.60 |
| *Qwen3 8B* | *GPT5 mini* | 0.99 | 0.45 |

Table 3: Performance across different models for $k = 5$ without changelog.

Table 2 shows the accuracy of binary diff summarization and malware detection with *GPT5 mini* as measured by precision ($P$) and recall ($R$) described in Section 4.2. We run the prediction step with four configurations by modifying $k$, the number of functions with highest FSS provided to the LLM, and whether or not the program changelog was provided. The changelog is extracted from the program source code or online repository for each version and appended to the initial prompt when needed. The gray highlighted rows show the overall $P$ and $R$, while the rows beneath them show per-program results. **Precision** ($P$) ranges from 0.94 to 0.96, indicating that the framework obtains low false positives. $P$ improves slightly with changelog for $k = 5$ but stays the same for $k = 10$. **Recall** ($R$) is highest at 0.71 for $k = 5$

without changelog. $R$ drops to 0.62–0.63 with $k = 10$, indicating that a larger context of functions may confuse the LLM prediction. Including changelog leads to a slight drop in $R$ for both $k$.

Across **programs**, $P$ and $R$ show wide variation. $P$ ranges from 0.88 to 1.00, with all other programs getting near perfect $P$ except `openssl` and `tar` where false positives are high, likely because of cryptographic operations and directory access. A similar dip in $R$ is seen for `openssl` and `tar`. $R$ ranges from 0.51 for `openssl` to 0.92 for `paho-mqtt`, indicating that the framework identifies malware better for certain programs than others.

Table 3 presents the results for the rest of the LLMs, along with combinations of weaker and stronger LLMs. We evaluate only one configuration of $k = 5$ without changelog. In our experiments, the *Qwen3* models work well when generating summaries but stumble in the final prediction step; in many cases they do not produce either MALICIOUS or BENIGN as instructed and a prediction is not obtained from their response. To overcome this, we use the *Qwen3* summaries and perform the final prediction step with *GPT5 mini* for these and other models. We notice *GPT OSS 20B* performs better than *GPT5 nano*. All the models perform better with *GPT5 mini* as the predictor, and *GPT5 mini* shows slight improvement in $P$ and $R$ when using *GPT5* as predictor. This indicates that summaries generated by all the models are of sufficient quality to allow for higher capability LLMs to perform accurate malware detection. With *GPT5 mini* as predictor, *GPT OSS 20B* outperforms *Qwen3* and *GPT5 nano*, fairing the best among lower capability models with $R$ of 0.63. *Qwen3 30B* comes second with $R$ of 0.60, followed by *Qwen3 8B* and *GPT5 nano*.

| Ablation | $P$ | $R$ |
|---|---|---|
| Full system | 0.95 | **0.71** |
| w/ diff size | 0.92 | 0.25 |
| w/ syscall | 0.97 | 0.57 |
| w/o diff callgraph | 0.97 | 0.48 |
| w/ threshold | 0.93 | 0.65 |
| `malcontent` (baseline) | 1.00 | 0.18 |

Table 4: Ablation study and baseline comparison with the full system as *GPT5 mini* for $k = 5$ without changelog.

**Ablation studies:** Table 4 shows the results of ablation studies to demonstrate the utility of each component of our framework. Configuration of *GPT5 mini* with $k = 5$ and no changelog is used for the ablations. To check the impact of FSS score on the final prediction, we replace FSS score based top-$k$ selection with two metrics: diff size in terms of number of lines (row "w/ diff size"), and number of system calls referenced by the function (row "w/ syscall"). These metrics are crude in contrast with FSS as they do not truly capture the sensitivity of code changes. With diff size, $R$ drops significantly to 0.25, as expected because larger diffs only highlight parts of the binary where most changes happened but these need not coincide with small malicious changes. With system calls, $R$ degrades to 0.57, indicating that even looking at certain behavior aspects via system calls is not sufficient as a high number of system calls is not truly indicative of sensitive behaviors. Additionally, this method will not scale as malwares may not invoke a large number of system calls for malicious behavior. $P$ drops slightly with diff size to 0.92 but improves with syscalls to 0.97.

We examine the impact of the diff callgraph by removing it (row "w/o diff callgraph") such that no dependency information is provided during summarization. $R$ drops to 0.48 however $P$ improves slightly to 0.97, indicating drop in false positives with increase in false negatives. This demonstrates that the diff callgraph and dependency information help produce higher quality summaries that helps downstream malware detection. Lastly, we remove the final LLM prediction step and use thresholds on FSS scores for malware detection (row "w/ threshold"). We use only the top FSS score and mark the binary as MALICIOUS if the FSS score is higher than a threshold. The threshold is calibrated using `gzip` to maintain $P \geq 0.95$, giving a threshold of 8.5. The rest of the dataset is evaluated to give $P$ 0.93 and $R$ 0.65. We see only a small drop in $R$, demonstrating that FSS scores produced in the summarization step are of high quality. However, even with a high threshold of 8.5, $P$ shrinks as false positives increase, indicating that this method may not scale to varied data where even benign functions would be marked with high sensitive score.

**Baseline:** Table 4 also show a comparison with the `malcontent` rule-based binary diff malware scanner ChainGuard (2025) which we treat as a baseline. `malcontent` looks at binary changes and marks each change with LOW, MEDIUM, HIGH, or CRITICAL severity. According to the instructions, CRITICAL should be treated as malware, however we consider both HIGH and CRITICAL as malware. `malcontent` faces a sharp decline in $R$ to 0.18, as its rule based scanner

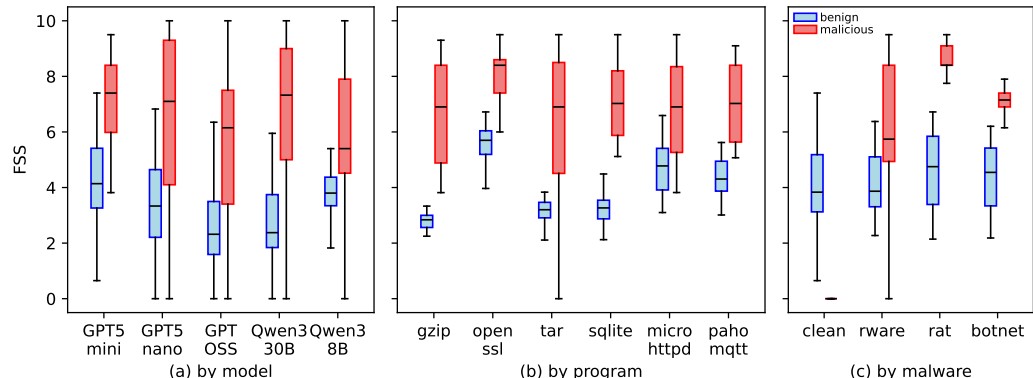

Figure 5: Distribution of $FSS_{\text{ben}}$ and $FSS_{\text{mal}}$ (a) by LLM, (b) by program for *GPT5 mini*, and (c) by malware for *GPT5 mini*. The boxes show first to third quartile, the middle line shows median, and the whiskers show $1.5\times$ inter-quartile range.

cannot pick up previously unseen malwares. It achieves perfect $P$, indicating that it is conservative in marking changes as HIGH or CRITICAL. Our framework improves $R$ by $\approx 4\times$ with only a small decline in $P$. Traditional rule-based cybersecurity tools require constant updates to the set of rules to detect new malwares, motivating the need for an LLM-driven binary diff summarization framework. Comparison with this baseline demonstrates that malware detection, especially of unseen malware, is a hard problem which our binary diff summarization framework tackles effectively to achieve a recall of 0.71.

**FSS Separation:** Figure 5 presents the distribution of FSS scores for benign ($FSS_{\text{ben}}$) and malicious ($FSS_{\text{mal}}$) functions as described in Section 4.2. Figure 5(a) shows distribution by model. All models demonstrate a clear separation between $FSS_{\text{ben}}$ and $FSS_{\text{mal}}$. Except for *GPT5 mini*, other models have a wider spread in the $FSS_{\text{mal}}$ distribution that overlaps with $FSS_{\text{ben}}$, yet the boxes and medians remain clearly separated. *GPT5 mini* shows a tighter distribution of $FSS_{\text{mal}}$ than the rest, demonstrating greater scoring consistency. Overall, the median scores show a difference of 1.5 to 5.0 points, with *GPT5 mini* having a separation of 3.0. The summarization framework reliably marks malicious injected functions with higher FSS than benign functions. The benign functions are consistently scored with median FSS of 4.0 or lower; LLM understands function sensitivity correctly, illustrating efficacy of FSS categorization.

To further investigate the performance of the best-performing model *GPT5 mini*, Figures 5(b) and 5(c) provide a granular breakdown of *GPT5 mini*'s scores. The analysis by program demonstrates that *GPT5 mini*'s discriminative power is robust across the set of programs. Interestingly, the $FSS_{\text{ben}}$ distributions across programs are narrow, showing that *GPT5 mini* consistently marks the functions similarly. Additionally, `openssl`, `microhttpd`, and `paho-mqtt` get higher $FSS_{\text{ben}}$ as expected because the benign functions have cryptographic and network functionalities. Similarly, Figure 5(c) illustrates the model's effectiveness against different malwares. Distribution of $FSS_{\text{mal}}$ for `rware` is largest, while for `rat` and `botnet` is very narrow, indicating that it is easier to identify sensitive behaviors with the network access in the later two. Nonetheless, there is a clear separation across all malwares which makes it easy to configure thresholds for detection.

**False Negative Analysis:** We compute the recall $R$ per malware (equivalent to true positive rate) for *GPT5 mini* with $k = 5$ without changelog to shine light onto the false negative cases. The overall $R$ is 0.71. Per malware, the $R$ is 0.76 for `rware`, 0.93 for `rat`, and 0.45 for `botnet`. `botnet` is significantly lower than the other two, displaying the framework's weakness in terms of detecting this type of malware. As the `botnet` only listens for a network connection and sends UDP packets, its malicious behavior might be harder to identify than the file encryption behavior of `rware` or reverse shell of `rat`.

## 6 CASE STUDY: XZ BACKDOOR

We analyze the XZ Utils supply chain attack detected in 2024 (Przymus & Durieux, 2025), where the open-source XZ repository was compromised to inject a backdoor into the `liblzma.so` library. This library is ubiquitous on Linux systems ranging from servers to embedded controllers, so the attack would have devastating consequences, however it was caught before the backdoor was distributed as part of updates. We compile the XZ utils source code for the compromised version `v5.6.0` and a previous version `v5.4.7`. We evaluate our binary diff summarization framework on the generated `liblzma.so` libraries. We run *GPT5 mini* and *GPT5* for both the summarization and prediction step with $k = 5$ and no changelog.

| | **Predictor** | | **Top-5 functions** |
|---|---|---|---|
| **Summarizer** | GPT5 | GPT5 mini | |
| GPT5 | ⚠️ malicious | ⚠️ malicious | FUN_00104794(6.5) FUN_00104720(6.5) _get_cpuid(5.3) lzma2_decode(4.3) FUN_0011e4a0(4.2) |
| GPT5 mini | ⚠️ malicious | ✅ benign | x86_code(4.2) FUN_00104794(4.2) crc64_set_fun(3.4) _get_cpuid(3.4) FUN_00104720(3.4) |

Table 5: XZ backdoor detection by *GPT5* and *GPT5 mini* along with sensitive functions.

Table 5 shows the output of malware detection by GPT5 and *GPT5 mini* when run on each other's summarizations. GPT5 correctly marks the diff summaries as malicious for both the summaries generated by itself and by *GPT5 mini*. On the other hand, *GPT5 mini* misclassifies its own summaries as benign, however it correctly marks GPT5 summaries as malicious. This indicates that both models highlight the injected malicious behavior sufficiently, while it takes the more capable GPT5 for a correct prediction.

The top-5 highest scored functions are shown along with their scores for both models. The red highlighted functions were those injected with malicious behavior. Out of 79 functions in the diff, both models score the relevant malicious functions higher so they appear among the top 5. GPT5 scores the malicious functions highest, whereas *GPT5 mini* scores them generally lower. This demonstrates that the LLMs correctly identify sensitive behaviors using the FSS categorization.

Figure 6 shows the diff summaries generated by GPT5 for the two highlighted functions. Highlighted in red, we see the model describe how the functionalities are "atypical for liblzma" and differ from "liblzma's expected functionality". This case study illustrates that LLMs utilize the binary diff summarization framework and FSS categorization to produce meaningful summaries that highlight malicious behavior when analyzing software updates.

**FUN_00104720 diff summary** 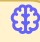

Newly added function. It introduces a call-count–gated side effect (only when the global counter equals 1 on entry) that invokes NEWF_00104794 with a crafted local buffer containing param_2. That callee performs an unchecked transient memory write to an address derived from pointer arithmetic and issues a CPUID call before restoring memory. This is **atypical for liblzma and poses potential integrity/availability risks** (fault or probe) and increases hardware fingerprinting/anti-analysis behaviors...

**FUN_00104794 diff summary** 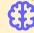

New function added...introduces opaque initialization and a transient write to an address computed from (vtable_pointer − this + 0x2e8), followed by a CPUID call and restoration... Unchecked computed-pointer dereference/write (may crash ... integrity/availability risk)... Stores pointer/offset deltas into object fields, **suggesting obfuscation or anti-reverse-engineering behavior rather than typical liblzma state setup. Given liblzma's expected functionality, this routine is anomalous and warrants deeper review for malicious or vulnerable behavior...**

Figure 6: GPT5 summaries for the XZ backdoor functions.

## 7 CONCLUSION

In this work, we presented a novel framework for binary diff summarization using LLMs, with a specific focus on enhancing software supply chain security. We introduce the functional sensitivity

score (FSS), a metric designed for automated triage of sensitive functions within binary diffs. To evaluate our approach, we created a new benchmark for software supply chain security, comprising 104 versions of 6 open-source projects, into which we injected 3 different types of malware. Our framework achieved a high precision of 0.95 and a recall of 0.71 for malware detection. Furthermore, the FSS demonstrated a clear separation of 3.0 points between malicious and benign functions, highlighting its effectiveness. On the real-world XZ backdoor case study, our framework correctly captured the injected malicious functions with high FSS and correctly marked the software update as malicious, exemplifying the applications to real-world scenarios. These findings illustrate the significant potential of leveraging LLMs for automation of software supply chain security. Future work could explore the application of this framework to other security-critical domains, such as vulnerability detection and patch analysis. The FSS could be adapted and refined for other security applications, and the framework could be extended to support a wider range of architectures.

**Ethics:** This work explores the use of large language models (LLMs) for binary diff summarization, which identifies changes between binary versions to help analysts detect bugs, vulnerabilities, and supply chain threats. Although the technique strengthens patch management and software integrity verification, it also has dual-use implications. Malicious actors could potentially exploit the same methods for reverse engineering, intellectual property theft, or scalable attacks on software supply chains. Our study is conducted purely for defensive and research purposes, aiming to advance the ability of the security community to manage patches and identify vulnerabilities. We acknowledge the risks of misuse and emphasize the importance of safeguards, rigorous evaluation, and governance mechanisms in guiding responsible adoption of LLM-based tools. By contextualizing and transparently reporting our findings, we seek to raise awareness of emerging attack vectors while supporting the development of effective countermeasures.

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

## A APPENDIX

### A.1 RELATED WORK COMPARISON

### A.2 BENCHMARK DETAILS

Table 7 lists the URLs for each open-source project in our software supply chain security benchmark.

### A.3 MALWARE IMPLEMENTATIONS DETAILS

**Ransomware:** The ransomware is implemented as a C program that utilizes self-contained versions of tiny-AES[2] and tiny-ECDH[3] for its cryptographic operations. The malware recursively scans for files and encrypts each one with a unique, randomly generated AES-128 key in Counter (CTR) mode, appending a .CRYPT extension to the filename. To protect these individual file keys, it employs an Elliptic Curve Diffie-Hellman (ECDH) key exchange using the NIST B-163 curve; it

---

[2]https://github.com/kokke/tiny-AES-c
[3]https://github.com/kokke/tiny-ECDH-c

| Tool / Paper | Category / Approach |
|---|---|
| BMAT (Wang et al., 2000) | Symbol / Name-based / Fuzzy |
| BinDiff (Flake, 2004) | Graph-based |
| BinDiff extended (Dullien & Rolles, 2005) | Graph-based |
| BinHunt (Gao et al., 2008) | Graph-based + Symbolic Execution |
| DarunGrim (Oh, 2008) | Graph-based |
| BinSlayer (Bourquin et al., 2013) | Graph-based + Bipartite matching |
| Diaphora (Koret, 2015–2025) | Graph-based |
| Asm2Vec (Ding et al., 2019) | ML-based embedding |
| DeepBinDiff (Duan et al., 2020) | ML-based embedding |
| jTrans (Wang et al., 2022) | Deep Learning |
| QBinDiff (Cohen et al., 2024) | Network alignment/Belief Propagation |

Table 6: Implementation approach for related works

| Program | URL |
|---|---|
| gzip | https://ftp.gnu.org/gnu/gzip |
| openssl | https://github.com/openssl/openssl/releases/download |
| tar | http://mirror.rit.edu/gnu/tar |
| sqlite | https://sqlite.org |
| microhttpd | https://ftp.gnu.org/gnu/libmicrohttpd |
| paho-mqtt | https://github.com/eclipse-paho/paho.mqtt.c/archive/refs/tags |

Table 7: URLs for each project in the benchmark.

generates a shared secret by combining a new local private key with a hardcoded attacker's public key. This shared secret is then used as a master key to encrypt all the individual file keys and their paths into an info.bin file, after which the ransomware drops a note containing the victim's public key needed for decryption.

**Remote access trojan:** The RAT implements a stealthy reverse shell that connects a target machine back to an attacker. It begins by reading the attacker's IP address and port from an environment variable, a technique used to avoid hardcoding sensitive information. The program then uses fork() to create a child process, allowing the parent to exit immediately while the malicious code continues to run in the background, detached from the original application. This child process establishes a network connection to the attacker's machine. The core of its functionality lies in using the dup2() system call to redirect the standard input, output, and error streams to the network socket. Finally, it calls execve() to replace its own process with /bin/sh, which is cleverly obfuscated in the code as a series of integer multiplications. Because the I/O streams are already redirected, this new shell process is fully interactive for the remote attacker, granting them command-line control over the compromised system.

**Botnet:** The botnet client is implemented based on the leaked source code of the Mirai botnet (Antonakakis et al., 2017). The program is designed to connect to a Command and Control (C2) server, which is hardcoded as "localhost" on port 5034. Once connected, the bot enters a loop where it sends a periodic keep-alive message to the C2 server and listens for attack commands. When a command is received, it is parsed to extract a target IP address, port, payload size, and the number of packets to send. Unlike the original Mirai, which featured multiple attack vectors, this simplified version only implements a basic UDP flood attack. This attack function bombards the specified target with a high volume of UDP packets containing randomized data, generated by a Xorshift pseudo-random number generator identical to the one used in Mirai, with the goal of overwhelming the target's network resources.

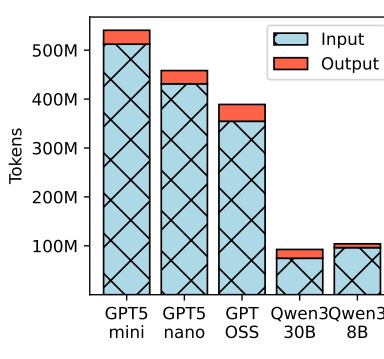

Figure 7: Token consumption.

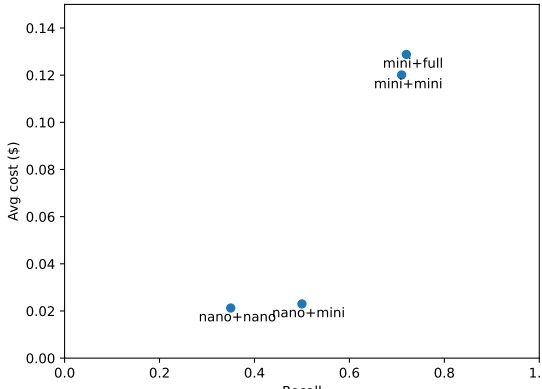

Figure 8: Cost vs Recall.

## A.4 TOKEN CONSUMPTION AND COST ANALYSIS

Figure 7 shows the total input and output token consumption per model on the entire benchmark. The tokens range from 100M for Qwen3 models to 500M for *GPT5 mini*. The wide difference in token consumption may be due to different tokenizers for each model and because *GPT5 nano* and *GPT5 mini* may produce larger and more detailed function summaries that are sent back in the followup prompt. Output tokens are around 5% to 25% of input tokens. Considering 46K functions in the benchmark, the average per-function token consumption is around 2K to 12K.

Figure 8 shows the cost versus recall analysis for different GPT5 variants. Each point is labeled with two models indicating the summarizer + predictor. *GPT5 mini* and *GPT5* predictors see a sharp increase in recall due to higher capabilties, however the cost only changes slightly. The recall increases sharply with more capable predictors. However, it is expected that recall will saturate with increasing cost.

