# OpenReview forum: "Binary Diff Summarization using Large Language Models"
_ICLR.cc/2026/Conference — Submitted to ICLR 2026_

### Official Review · Reviewer_kc2V · 2025-10-28

**Soundness:** 1
**Presentation:** 3
**Contribution:** 2
**Rating:** 2
**Confidence:** 5

**Summary:**

This paper addresses the challenge of detecting malicious or vulnerable changes in software supply chain updates by proposing a binary diff summarization framework using LLMs. The framework combines traditional binary diffing with natural language summarization and introduces the Functional Sensitivity Score (FSS), a metric inspired by CVSS, to triage binary functions according to sensitivity. The paper creates a benchmark of six open-source projects injected with three types of malware, totaling 104 versions, 392 binary diffs, and over 46K functions. Experiment results show high precision and reasonable recall in malware detection, with clear separation between benign and malicious functions.

**Strengths:**

1. Introduces an integration of binary diffing, LLM summarization, and FSS scoring, enabling triage of sensitive code changes.
2. Builds a realistic dataset by injecting ransomware, RAT, and botnet malware into six widely used open-source projects.

**Weaknesses:**

1. The paper offers limited technical novelty.
2. The experiments are not comprehensive.

**Questions:**

This paper introduces an interesting pipeline, but in its current form it is not strong enough for acceptance.

1. The overall approach (diff → LLM summary → score) is primarily a pragmatic composition rather than a new technical contribution. Since FSS is directly adapted from CVSS with hand-tuned weights and the final malicious/benign decision is delegated to another LLM, the novelty is insufficient.

2. There are many relevant baselines (e.g., those in Appendix A.1 such as DeepBinDiff). How does the proposed method compare against these in terms of accuracy and efficiency? In addition, how do different prompting strategies (e.g., zero-shot vs. chain-of-thought vs. few-shot) or reasoning-optimized/domain-specific models affect performance?

3. There are several existing datasets, such as those from DeepBinDiff or BinSimDB[1], etc. They could provide stronger external validation. Why were these not used, and how does the proposed dataset in this paper differ from or improve upon them? And how's the performance of the proposed framework on those datasets?

4. The framework does not clearly define a threshold for distinguishing malicious from benign using the FSS. For a deployable tool, a principled thresholding or calibration strategy seems necessary.

5. The results show significant variance in recall across models and programs. Could the paper provide a detailed breakdown of failure cases to illustrate where and why the framework fails?

Reference:

[1] Zuo, Fei, Cody Tompkins, Qiang Zeng, Lannan Luo, Yung Ryn Choe, and Junghwan Rhee. "BinSimDB: Benchmark Dataset Construction for Fine-Grained Binary Code Similarity Analysis." In International Conference on Security and Privacy in Communication Systems, pp. 203-225. Cham: Springer Nature Switzerland, 2024.

---

> ### Author Response · Authors · 2025-11-22
>
> We thank you for your comments and feedback. We have responded to the weaknesses and questions below.
>
> > 1. The overall approach (diff → LLM summary → score) is primarily a pragmatic composition rather than a new technical contribution. Since FSS is directly adapted from CVSS with hand-tuned weights and the final malicious/benign decision is delegated to another LLM, the novelty is insufficient.
>
> The novelty of our approach lies in bridging the gap between existing binary diffing tools and automated triage of software updates for software supply chain security. While existing tools excel at identifying syntactic changes, our binary diff summarization method takes that ahead by constructing a diff callgraph, summarizing each change, categorizing functional sensitivity using FSS scoring, and performing malware detection by prediction on the top-scored functions. A direct combination of LLMs with binary diffing tools may face “needle in a haystack” problems where the LLM cannot reliably isolate malicious parts of the diff. Our framework’s novelty lies in tackling this problem using a principled iterative approach of constructing and traversing the diff callgraph to isolate sensitive function behaviors for downstream triage. Additionally, FSS is not directly adapted from CVSS but merely inspired by it as the categories of CVSS are fundamentally different than what is required to classify sensitive behaviors and CVSS can not be used as a replacement of FSS.
>
> > 2. There are many relevant baselines (e.g., those in Appendix A.1 such as DeepBinDiff). How does the proposed method compare against these in terms of accuracy and efficiency?
>
> The related works presented in Section 2 and Appendix A.1 are not direct baselines as the prior works are primarily focused on the binary code similarity problem (i.e. matching similar binary functions) which is different than the binary diff summarization and malware detection problem that our framework addresses. While this aspect is important, it is incorporated into Ghidriff as the “correlations” mentioned in Section 3. Our binary diff summarization pipeline starts after a tool like Ghidriff or DeepBinDiff have adequately matched similar binary functions. Our framework is extensible to incorporate better binary code similarity tools, however we currently experiment only with Ghidriff as it is the only tool that directly provides decompiled code.
>
> > In addition, how do different prompting strategies (e.g., zero-shot vs. chain-of-thought vs. few-shot) or reasoning-optimized/domain-specific models affect performance?
>
> All models that we evaluate with are reasoning-optimized and reasoning is turned on for our experiments. While cybersecurity domain-specific models may perform better, the existing cybersecurity LLMs such as SecGemini and Lily-Cybersecurity-7B are outperformed by Qwen3 on cybersecurity baselines, hence we evaluate with Qwen3.
>
> > 3. There are several existing datasets, such as those from DeepBinDiff or BinSimDB[1], etc. They could provide stronger external validation. Why were these not used, and how does the proposed dataset in this paper differ from or improve upon them? And how's the performance of the proposed framework on those datasets?
>
> As emphasized above, DeepBinDiff and BinSimDB target a different problem of binary code similarity, hence their datasets are tailored for that task. Their datasets are not tailored to software supply chain security and do not contain any malware injected samples which could allow us to evaluate malware detection. This is why we choose to create our own dataset.
>
> > 4. The framework does not clearly define a threshold for distinguishing malicious from benign using the FSS. For a deployable tool, a principled thresholding or calibration strategy seems necessary.
>
> We clarify that our complete framework integrates a final LLM predictor that directly outputs the prediction label, so that such threshold calibration need not be performed. An LLM can make the final prediction by reasoning about the highest scored functions. FSS by itself is designed to classify sensitive behavior, which may give high scores for sensitive but benign binary diffs that affect critical parts of the code. Nonetheless, we have included a threshold based prediction as part of our ablation study described in Section 5. We calibrate the threshold on gzip to maintain $P \geq 0.95$, obtaining a threshold of 8.5. We evaluate this threshold on the remaining dataset to get P as 0.93 and R as 0.65. Such a high threshold value may not scale well to different types of malware, especially those that may have less sensitive behaviors.
>
> (Continued)

---

> ### Author Response · Authors · 2025-11-22
>
> > 5. The results show significant variance in recall across models and programs. Could the paper provide a detailed breakdown of failure cases to illustrate where and why the framework fails?
>
> We analyzed the recall per malware type (which is equal to true positive rate) and found that `botnet` gets the least recall of 0.45 whereas `rware` gets 0.76 and `rat` gets 0.93. This indicates that identifying the botnet as malicious is relatively challenging, likely because the botnet only listens for network access and sends UDP packets. These behaviors are not, by themselves, necessarily malicious. Hence, in the context of balancing the false positive and false negative aspects, it is seen that it is likely that the botnet is not marked as malicious.  We have added these details to Section 5.

---

### Official Review · Reviewer_qTpZ · 2025-10-29

**Soundness:** 3
**Presentation:** 3
**Contribution:** 2
**Rating:** 4
**Confidence:** 2

**Summary:**

The author(s) propose a novel framework that combines LLM-based binary code summarization with binary diffing to enhance the model's ability to focus on critical code changes and support complex tasks such as automated malware detection.

**Strengths:**

They create a software supply chain security benchmark by injecting 3 different malware into 6 open-source projects which generate 104 binary versions, 392 binary diffs, and 46,023 functions.

**Weaknesses:**

1. The methodological justification requires more clarity. It would be beneficial if the authors could explain why the proposed combination of LLM summarization and binary diffing offers advantages over existing binary analysis or summarization approaches.
2. The experimental section needs more comparisons with related baselines.
3. The paper would benefit from a more detailed discussion of the hyperparameter settings and ablation study used during the framework and evaluation.

**Questions:**

see Weaknesses.

---

> ### Author Response · Authors · 2025-11-22
>
> We thank you for your comments and feedback. We have responded to the weaknesses and questions below.
>
> > 1. The methodological justification requires more clarity. It would be beneficial if the authors could explain why the proposed combination of LLM summarization and binary diffing offers advantages over existing binary analysis or summarization approaches.
>
> We have revised the Introduction and Related Work section to emphasize the motivation of our framework. Traditional binary diffing tools (e.g., BinDiff, Diaphora) rely on syntactic and graph-based similarity to identify changed components. While effective at pinpointing where changes occur, they require significant manual effort and domain expertise to interpret. Our framework bridges this gap by overlaying LLM-generated natural language summaries onto these diffs, allowing analysts to instantly grasp the semantic meaning of the changes rather than just the syntactic differences (as exemplified by the XZ case study). Additionally, standard diffing tools cannot assess the "risk" of a change, whereas our pipeline uses the semantic understanding from the LLM to score functions using FSS. This provides an automated triage mechanism that isolates high-risk changes for human review, a capability not present in existing summarization or diffing tools alone.
>
> > 2. The experimental section needs more comparisons with related baselines.
>
> We have added a comparison with the “malcontent” diff-based malware scanner as baseline. “Malcontent” achieves a precision of 1.00 and recall of 0.18, displaying a significant drop in malware detection accuracy compared to our framework.
>
> > 3. The paper would benefit from a more detailed discussion of the hyperparameter settings and ablation study used during the framework and evaluation.
>
> Our study already included the impact of hyperparameters $k$ and providing changelog for prediction.
>
> We have performed ablation studies with the following cases:
> Remove FSS scores and use diff size to rank functions for prediction
> Remove FSS scores and use count of syscalls to rank functions for prediction
> Remove prediction step and use threshold on FSS
> Remove diff callgraph and dependencies for summarization
> The table below shows that replacing each component of our framework leads to performance degradation, thus strengthening our framework’s design choice. We have included the table and details of the ablation study in Section 5.
>
> |Ablation|P|R|
> |---|---|---|
> |Full system|0.95|**0.71**|
> |diff size|0.92|0.25|
> |syscall|0.97|0.57|
> |threshold|0.93|0.65|
> |w/o diff callgraph|||
> |---|---|---|
> |malcontent (baseline)|1.00|0.18|

---

### Official Review · Reviewer_KHs7 · 2025-10-30

**Soundness:** 2
**Presentation:** 2
**Contribution:** 2
**Rating:** 4
**Confidence:** 4

**Summary:**

The paper proposes a framework for binary diff summarization using LLMs. Given two binary versions, the system uses Ghidriff to extract only the added, deleted, and modified functions, constructs a diff callgraph, and then processes functions in reverse BFS order so that each function's LLM prompt can include summaries of its callees. On top of the summaries, the paper introduces a Functional Sensitivity Score, inspired by CVSS, with five categories
$B,R,C,I,A \in \{\text{none},\text{low},\text{medium},\text{high}\},$
aggregated as
$S = 1 - (1 - B)(1 - R), \quad M = 1 - (1 - C)(1 - I)(1 - A),$
$\text{FSS} =
\begin{cases}
\text{roundup}(5.3S + 6.1M), & M > 0,\\
0, & \text{otherwise.}
\end{cases}$
Then the top-$k$ highest-FSS functions are fed to the LLM again to classify the whole diff as MALICIOUS or BENIGN. The authors also build a supply-chain benchmark (6 OSS projects, 104 versions, 392 diffs, 46023 functions, 3 malware families). With GPT-5 mini and $k=5$ + changelog, they report precision 0.98 and recall 0.64.

**Strengths:**

Originality: Instead of plain binary summarization, the method couples diffing, callgraph-aware prompting, and LLM-based triage (FSS), which is a novel combination for supply-chain inspection.

Quality: FSS is fully specified with explicit weights and formulas, and the two-prompt LLM pattern (summary $\to$ FSS) is clearly shown (Fig. 2).

Clarity: The pipeline (Fig.~1) is easy to follow and reverse-BFS ordering is well motivated.

Significance: The dataset is sizable and realistic for OSS supply-chain updates and the XZ case study connects the framework to a real incident.

**Weaknesses:**

Mid recall. The best setting gets precision 0.98 but recall only 0.64, meaning ~36% of malicious updates are missed. This is high precision, not high accuracy. A threshold/$k$ sweep is needed.

LLM brittleness. Some models do not output the requested labels, and the paper fixes this by using GPT-5 mini as the final predictor for all models (Table 3). This shows the method depends on a particular LLM.

FSS validation is indirect. FSS is evaluated against the very malware the authors injected (malicious vs. benign functions), not against human-labeled sensitivity or a trivial rule baseline.

Benchmark scope. All malwares are source-level, compiled with the same toolchain, and there is no obfuscation/packing, so the claim should be narrowed to "LLM-assisted analysis of cleanly injected updates.''

No FSS ablation. There is no experiment showing that FSS is actually better than, e.g., selecting functions by the presence of network/crypto/file ops.

**Questions:**

1- Can you report precision/recall for more $k$ values and/or FSS thresholds so we can see whether recall can be raised above $0.8$ at acceptable precision?

2- On what data were the FSS weights/tuning done, and were the same weights used on the XZ case? Please clarify to rule out test-set tuning.

3- In the XZ case, GPT-5 mini misclassifies its own summaries but GPT-5 succeeds. Does this mean the final predictor must be a high-capability LLM?

4- What happens when Ghidriff mismatches or fails to match a modified function? Is the downstream LLM step robust to an incomplete diff callgraph?

5- Will you release sanitized ghidriff outputs + prompts + LLM responses so others can test the FSS part without distributing malware binaries?

---

> ### Author Response · Authors · 2025-11-22
>
> We thank you for your comments and feedback. We have responded to the weaknesses and questions below.
>
> > Mid recall. The best setting gets precision 0.98 but recall only 0.64, meaning ~36% of malicious updates are missed. This is high precision, not high accuracy. A threshold/ sweep is needed.
> > 1- Can you report precision/recall for more  values and/or FSS thresholds so we can see whether recall can be raised above  at acceptable precision?
>
> We have improved the summarization and prediction prompts slightly to include snippets of the diff with the summary, while keeping the rest of the pipeline the same. With this, GPT5 mini achieves a recall of 0.71 and precision of 0.95. We reiterate that the recall indicates accuracy of malware detection of previously unseen malwares in a zero-shot setting as we do not perform training or in-context learning of the LLMs. In this setting, our framework achieves a 71% malware detection accuracy. This is quite high for the zero-shot setting with unseen malwares. This is also evident when comparing with a rule-based scanner baseline “malcontent” that achieves a recall of 0.18. Details of the baseline comparison have been added to Section 5.
>
> Our framework does not utilize FSS thresholding as the final LLM predictor directly generates a malicious or benign label, thus a threshold sweep is not possible for the full framework. Nonetheless, we have added an ablation study that utilizes FSS thresholding instead of the LLM prediction, achieving 0.93 precision and 0.65 recall. Details of the ablation studies have been added to Section 5.
>
> > Benchmark scope. All malwares are source-level, compiled with the same toolchain, and there is no obfuscation/packing, so the claim should be narrowed to "LLM-assisted analysis of cleanly injected updates.''
>
> We argue on the contrary that source-level malwares do not remain cleanly injected in the binary as the process of compilation acts as a natural obfuscation. At the binary level, cleanly injected malwares would not modify the rest of the binary code and only inject additional malware-related code (e.g. via dynamically loaded libraries), whereas a source level malware would affect the entire binary due to compiler optimizations. Additionally, we consider the same compilation toolchain as encountering binaries compiled with different toolchains in a software supply chain scenario can be considered as anomalous from the viewpoint of software supply chains, especially for embedded devices which are a main focus in this study.
>
> > LLM brittleness. Some models do not output the requested labels, and the paper fixes this by using GPT-5 mini as the final predictor for all models (Table 3). This shows the method depends on a particular LLM.
>
> We clarify that for summarization which is the core of our framework, Qwen3 models perform adequately well as seen in Table 3 and Figure 5(a). Qwen3 models only stumble during the final prediction step. On further analysis, we observed that this happens due to context length limitations that were frequently exceeded when passing multiple function summaries for the final prediction. The context length limitations came from the model and local hardware restrictions.
>
> > FSS validation is indirect. FSS is evaluated against the very malware the authors injected (malicious vs. benign functions), not against human-labeled sensitivity or a trivial rule baseline.
> > No FSS ablation. There is no experiment showing that FSS is actually better than, e.g., selecting functions by the presence of network/crypto/file ops.
>
> We have performed two ablations where instead of using FSS to select top-$k$ functions for prediction, we utilize two other metrics: (1) size of diff in number of lines, (2) count of system calls (network, file) referenced by the function. Both these ablations lead to a drop in precision and recall, indicating that FSS helps improve malware detection performance by identifying sensitive functions. Human-labelling of the entire dataset would require substantial effort. Instead we demonstrate performance of an application of FSS, while also providing a visualization of FSS distribution and ablation studies. Together, these strengthen our claim that FSS is indeed useful in the context of binary diff summarization for malware detection.
>
> > 2- On what data were the FSS weights/tuning done, and were the same weights used on the XZ case? Please clarify to rule out test-set tuning.
>
> FSS weight tuning was not performed on any data, so there was no test-set tuning. Weights were tuned such that the FSS and CVSS curves matched and covered the 0 to 10 range when plotting all possible selections of categories. The same FSS weights were used for the XZ case study.
>
> (Continued)

---

> ### Author Response · Authors · 2025-11-22
>
> > 3- In the XZ case, GPT-5 mini misclassifies its own summaries but GPT-5 succeeds. Does this mean the final predictor must be a high-capability LLM?
>
> We provide results with GPT-5 as final predictor on GPT5 mini summaries. GPT-5 achieves precision 0.97 and recall 0.72, only slightly better than GPT5 mini. While the XZ case shows that higher capability LLM produces better predictions, the results on the dataset show that GPT5 mini suffices. Additionally, GPT5 mini helps improve performance for lower capability models, indicating that higher capability LLMs for predictor help improve performance.
>
> > 4- What happens when Ghidriff mismatches or fails to match a modified function? Is the downstream LLM step robust to an incomplete diff callgraph?
>
> If a function is mismatched or fails to match by Ghidriff, it will show up as added or modified and the LLM summarization step will still see its code. For example, if a modified function is not correctly matched with the corresponding old function, it will show up as an added function. The LLM summary will still pick up on sensitive functionality and assign a high FSS. In this manner, the downstream LLM pipeline is robust to mistakes in the diff algorithm
>
> > 5- Will you release sanitized ghidriff outputs + prompts + LLM responses so others can test the FSS part without distributing malware binaries?
>
> As part of the supplementary material, we have already provided ghidriff outputs, LLM-generated summaries, LLM-generated predictions, along with prompts without including malware binaries. We will update these with data from the additional experiments performed during revision.

---

> > ### Comment · Reviewer_KHs7 · 2025-11-27
> >
> > Thanks for the detailed rebuttal and new ablations. I will maintain my evaluation scores.

---

### Official Review · Reviewer_2Hkg · 2025-11-02

**Soundness:** 2
**Presentation:** 3
**Contribution:** 3
**Rating:** 6
**Confidence:** 3

**Summary:**

The paper introduces an LLM-based binary diff summarization that can be used to detect maliciously injected code or vulnerabilities from software updates. This approach utilizes LLM to analyze binary code and diff. The paper introduces a software supply chain security benchmark with 3 malware injections into 6 open-source projects. The proposed LLM-based binary diff summarization technique is used to create summaries that a downstream detector (utilizing functional sensitivity scores) uses to detect the injected malware. The proposed techniques achieve a 98% precision and 64% recall on the introduced dataset.

**Strengths:**

+ Detecting malware is an important topic. Having a reliable automatic technique that can flag potential malicious changes would make an IT admin’s job much easier.
+ The precision is quite good without many false negatives.

**Weaknesses:**

- The paper did not discuss any potential baseline, nor did it compare to any baseline. It would be possible to use prior work in binary analysis to create a baseline.
- The recall is quite low at just under 60%. Having some quality analysis on modes of failure for those false negatives would give future researchers some insights into how to improve malicious diffs.
- The lack of an ablation study makes it hard to know how each component contributes to the overall effectiveness of the technique.

**Questions:**

1. What is the reason for not having a baseline? Can any of the prior binary analysis tools be used as a baseline?
2. Since FSS behaves similarly to CVSS3.1, can CVSS3.1 be used instead and still produce similar results?
3. Since using GPT5mini as a predictor while using a smaller model for summary produces very close results to the full GPT5mini stack. What are the differences in cost? Would a cost vs effectiveness analysis be done to show the cost vs gain?
4. What are the typical modes of failure for false negatives?

---

> ### Author Response · Authors · 2025-11-22
>
> We thank you for your comments and feedback. We have responded to the weaknesses and questions below.
>
> > 1. “The paper did not discuss any potential baseline…”, “ Can any of the prior binary analysis tools be used as a baseline…”
>
> We have added a comparison with the [malcontent](https://github.com/chainguard-dev/malcontent) tool as a baseline (see table at bottom). Malcontent is a YARA rule based malware scanner that operates on binary diffs. Malcontent achieves significantly lower recall of 0.18, which is expected as rule based scanners cannot detect previously unseen malwares unless the rules are updated. This further motivates our LLM-driven framework that can detect unseen malwares via binary diff summarization. We emphasize that prior research in binary differential analysis has focused on the task of binary code similarity. A novelty of our work is that we are the first to introduce an LLM-driven binary diff summarization framework with applications of malware detection, thus related works listed in Section 2 and Appendix A.1 cannot be evaluated as baselines because they target a different task.
>
> > The recall is quite low at just under 60%.
>
> We have improved the summarization and prediction prompts slightly to include snippets of the diff with the summary, while keeping the rest of the pipeline the same. With this, GPT5 mini achieves a recall of 0.71 and precision of 0.95. We reiterate that the recall indicates accuracy of malware detection of previously unseen malwares in a zero-shot setting as we do not perform training or in-context learning of the LLMs. In this setting, our framework achieves a 71% malware detection accuracy, which is quite high for the zero-shot setting with unseen malwares.
>
> > The lack of an ablation study…
>
> We have performed ablation studies with the following cases:
> Remove FSS scores and use diff size to rank functions for prediction
> Remove FSS scores and use count of syscalls to rank functions for prediction
> Remove prediction step and use threshold on FSS
> Remove diff callgraph and dependencies for summarization
> The table below shows that replacing each component of our framework leads to performance degradation, thus strengthening our framework’s design choice. We have included the table and details of the ablation study in Section 5.
>
> > 2. …can CVSS3.1 be used instead and still produce similar results?”
>
> No. CVSS is designed to score exploitable vulnerabilities and not sensitive function behaviors, thus it cannot be directly used in place of FSS. The categories and classifications of CVSS3.1 are fundamentally different. For instance, one of the categories in CVSS3.1 is Attack Vector which classifies how a vulnerability will be attacked (Network, Adjacent, Local, or Physical). This categorization has no direct bearing to classifying sensitive behaviors in code. Even though FSS is inspired by and has some overlap with CVSS, FSS fulfills a distinct purpose that is not met by CVSS and CVSS can not be used as a replacement for FSS.
>
> > 3. …What are the differences in cost? Would a cost vs effectiveness analysis be done to show the cost vs gain?...
>
> We have added a cost-vs-recall plot in the Appendix A.4 for the OpenAI models using the average API cost per diff in US$ to provide an effectiveness analysis. We have not included the open-source models as it is difficult to estimate the US$ cost of open-source models in comparable terms to the commercial models. Figure 7 already provided an alternate comparison in terms of tokens consumed.
>
> |Models|Avg. Cost (US$)|Recall|
> |---|---|---|
> |GPT5 nano+nano|0.021|0.35|
> |GPT5 nano+mini|0.023|0.50|
> |GPT5 mini+mini|0.120|0.71|
> |GPT5 mini+GPT5|0.129|0.72|
>
> > 4. What are the typical modes of failure for false negatives?, Having some quality analysis on modes of failure…,
>
> We analyzed the recall per malware type (which is equal to true positive rate) and found that `botnet` gets the least recall of 0.45 whereas `rware` gets 0.76 and `rat` gets 0.93. This indicates that identifying the botnet as malicious is relatively challenging, likely because the botnet only listens for network access and sends UDP packets. These behaviors are not, by themselves, necessarily malicious. Hence, in the context of balancing the false positive and false negative aspects, it is seen that it is likely that the botnet is not marked as malicious.  We have added these details to Section 5.
>
>
> Table for ablation studies and baseline comparison. Ablations are performed with GPT5 mini using $k=5$ without changelog.
>
> |Ablation|P|R|
> |---|---|---|
> |Full system|0.95|**0.71**|
> |diff size|0.92|0.25|
> |syscall|0.97|0.57|
> |w/o diff callgraph|0.97|0.48|
> |threshold|0.93|0.65|
> |malcontent (baseline)|1.00|0.18|

---

### Meta-Review · Area_Chair_1581 · 2026-01-02

**Summary:**

This paper proposes an LLM-assisted framework for binary diff summarization aimed at software supply chain security. The framework combines traditional binary diffing and callgraph analysis with LLM-based natural language summarization and a hand-crafted Functional Sensitivity Score (FSS), and applies a second LLM stage to classify diffs as malicious or benign. Reviewers agree that the application domain is important and that the pipeline is clearly described, and they appreciate the construction of a malware-injected benchmark and a real-world case study (XZ utils). However, the overall assessment is mixed to negative. While the rebuttal adds a rule-based baseline, ablation studies, and cost–recall analysis, the core concerns regarding limited technical novelty, incomplete and arguably weak baselines, mid-range recall, and heavy dependence on specific LLM behaviors remain. In particular, the work is viewed as a pragmatic system composition rather than a clear methodological advance, and the empirical gains, while non-trivial, are not sufficiently convincing to outweigh these concerns. As a result, the paper does not meet the bar for acceptance at ICLR.

**Reviewer Concerns:**

Across reviews, there is consensus that malware detection and supply-chain security are important problems, and that combining binary diffing with LLM-based summarization is a reasonable engineering direction. Reviewers also acknowledge that FSS is explicitly defined and that the pipeline is easy to follow. The rebuttal addresses several surface-level issues by adding a comparison to a rule-based scanner (malcontent), reporting ablations that remove individual components (e.g., FSS, callgraph context), providing a cost–recall analysis for different LLM configurations, and offering a qualitative discussion of false negatives by malware type.


Nevertheless, several critical concerns remain unresolved. First, technical novelty is limited: many reviewers view the framework primarily as a pragmatic composition of existing tools (binary diffing, LLM summarization, heuristic scoring inspired by CVSS, and a final LLM classifier), without introducing a new algorithmic principle or learning formulation. The rebuttal reiterates the “bridging the gap” argument but does not fundamentally strengthen the novelty claim. Second, baseline evaluation is weak: aside from a rule-based scanner with predictably low recall, there is no convincing comparison against stronger or more relevant prior approaches in binary analysis or malware detection, and the justification that related methods “target a different task” is not fully persuasive to reviewers. Third, effectiveness remains questionable: even after rebuttal, recall remains around 0.7 in a controlled, zero-shot setting, meaning a substantial fraction of malicious updates are missed, and the framework’s dependence on specific LLMs (e.g., GPT-5 mini as a final predictor) raises concerns about brittleness and reproducibility. Finally, validation of FSS is indirect: FSS is evaluated primarily on the same injected malware used to construct the dataset, without human-labeled sensitivity ground truth or broader external validation, and its claimed advantage over simpler heuristics remains only partially supported despite added ablations. Taken together, these issues limit confidence in the contribution’s generality and research depth.

**Reviewer Scores:**

Reviewer scores are polarized but lean negative overall, with several reviewers at or below the acceptance threshold and only one marginally above threshold while explicitly stating that rejection would also be reasonable. The rebuttal does not appear to have shifted reviewer opinions substantially; at least one reviewer explicitly maintained their original score after reviewing the additional experiments and clarifications. While some reviewers acknowledge incremental improvements and practical value, the dominant concerns regarding novelty, baseline strength, and robustness persist. Under a full discussion period, it is unlikely that these concerns would be resolved sufficiently to produce consensus toward acceptance. Therefore, the most plausible outcome remains rejection rather than convergence to a weak accept.

---

### Decision · Program_Chairs · 2026-01-26

Reject